# Effects of Nitrogen Supply on Induced Defense in Maize (*Zea mays*) against Fall Armyworm (*Spodoptera frugiperda*)

**DOI:** 10.3390/ijms231810457

**Published:** 2022-09-09

**Authors:** Wenxin Wang, Xiaoyi Wang, Huimin Liao, Yuanjiao Feng, Yeshan Guo, Yinghua Shu, Jianwu Wang

**Affiliations:** 1Key Laboratory of Agro-Environment in the Tropics, Ministry of Agriculture, South China Agricultural University, Guangzhou 510642, China; 2Guangdong Provincial Key Laboratory of Eco-Circular Agriculture, South China Agricultural University, Guangzhou 510642, China; 3Guangdong Engineering Research Centre for Modern Eco-Agriculture, Guangzhou 510642, China; 4Department of Ecology, College of Natural Resources and Environment, South China Agricultural University, Guangzhou 510642, China

**Keywords:** nitrogen supply, induced defense, *Zea mays*, *Spodoptera frugiperda*, plant volatiles

## Abstract

How nitrogen (N) supply affects the induced defense of plants remains poorly understood. Here, we investigated the impacts of N supply on the defense induced in maize (*Zea mays*) against the fall armyworm (*Spodoptera frugiperda*). In the absence of herbivore attack or exogenous jasmonic acid (JA) application, N supply increased plant biomass and enhanced maize nutrient (soluble sugar and amino acid) contents and leaf area fed by *S. frugiperda* (the feeding leaf area of *S. frugiperda* larvae in maize supplemented with 52.2 and 156.6 mg/kg of N was 4.08 and 3.83 times that of the control, respectively). When coupled with herbivore attack or JA application, maize supplemented with 52.2 mg/kg of N showed an increased susceptibility to pests, while the maize supplemented with 156.6 mg/kg of N showed an improved defense against pests. The changes in the levels of nutrients, and the emissions of volatile organic compounds (VOCs) caused by N supply could explain the above opposite induced defense in maize. Compared with herbivore attack treatment, JA application enhanced the insect resistance in maize supplemented with 156.6 mg/kg of N more intensely, mainly reflecting a smaller feeding leaf area, which was due to indole emission and two upregulated defensive genes, *MPI* (maize proteinase inhibitor) and *PAL* (phenylalanine ammonia-lyase). Hence, the optimal N level and appropriate JA application can enhance plant-induced defense against pests.

## 1. Introduction

To avoid being consumed by herbivorous insects, plants have evolved sophisticated mechanisms to trigger defense mechanisms [1]. Plants are able to synthesize structurally diverse secondary metabolites, many of which are toxic, repellent, or anti-digestive for insects [2,3]. In addition, plants emit volatile organic compounds (VOCs) to repel pests or attract the natural enemies of pests [4]. Plant defenses are mediated by phytohormones, including jasmonic acid (JA), abscisic acid (ABA) and salicylic acid (SA) [2,5,6,7,8]. However, plant defense against pests is considerably affected by environmental factors, including mineral nutrition [9,10,11]. These exogenous factors strengthen or weaken plant defense and result in better or worse resistance against pests [12,13].

Nitrogen (N) is among the main environmental factors affecting plant development, physiology and metabolism [9,14]. N assimilation is related to key plant physiological or metabolic processes, such as photosynthesis, photorespiration, respiration, amino acid synthesis and the tricarboxylic acid (TCA) cycle [15,16,17]. N also impacts the ability of plants to cope with biotic stress, including herbivore attack [18]. N fertilizer may not affect insect biology directly but change the host–plant morphology, biochemistry and physiology, which can improve nutritional conditions for herbivores [19]. Therefore, N supply has bottom-up effects on the ecological fitness of herbivores, including the selection of host plants, survival, growth, development, fecundity and population dynamics [10,20]. For example, N application caused rice to be susceptible to *Nilaparvata lugens* [19]. The increased N availability in maize leads to an increased preference for beet armyworm *Spodoptera exigua* [12] and the number of egg masses deposited by *Ostrinia furnacalis* [20].

The nutrient composition and secondary metabolites of plants varies with different N concentrations [13,19,21,22,23]. N inputs increased the soluble protein content in maize, tobacco (*Nicotiana attenuate*) and rice (*Oryza sativa*) seedlings [13,21,24]. However, the content of soluble sugar in alfalfa (*Medicago sativa*) leaves decreased with an increasing N supply [25]. N supply increased total phenols in maize [13] and total flavonoids in *Hypericum perforatum* and in leafy Brassica [26]. N supply altered the expression of crop constitutive defense genes and regulated secondary compound metabolism [22,27,28]. 

N supply also affects the emissions of VOCs from plants. Gouinguené & Turlings reported that unfertilized maize plants release significantly lower amounts of herbivore-induced VOCs than fertilized plants [29]. Under high N conditions, pest attacks significantly increased the contents of β-menthene and iso-limonene in tomato [30]. Under low N conditions, the cotton attack by *S. exigua* significantly increased the total amount and types of VOCs, such as (Z)-3-hexenyl, (*E*)-2-hexenal, β-caryophyllene, (E)-β-farnesene, α-humulene and γ-bisabolene [27,31,32]. N deficiency increased volicitin-induced volatile emission in maize [31]. However, an increase in N supply significantly reduced *S. exigua* larval-induced VOC concentrations in cotton [27,31,32]. 

Maize (*Zea mays*), one of the most important food crops, is cultivated around the globe, with a production of more than 1.14 billion tons in 2018 [2], and large yield losses are caused by insect injury [33]. The fall armyworm, *Spodoptera frugiperda* (Lepidoptera: Noctuidae), is one of the most devastating pests of maize in recent years and has spread rapidly to 26 provinces in China since December 2018 [34]. For approximately three decades, numerous laboratories have investigated the defense systems of maize against herbivores, including secondary metabolites [35,36,37], phytohormones [38] and the elicitation of VOCs [4]. However, the effects of environmental factors (e.g., N supply) on maize-induced defense patterns and its consequence for insect herbivores are rarely studied [12,13]. This study aimed to elucidate how N supply levels affect nutrients, secondary metabolites, phytohormones and VOCs emission in maize, which ultimately influence the induced defense in maize against the invasive pest *S. frugiperda*.

## 2. Results

### 2.1. Effects of N Supply on Maize Aboveground Biomass

N supply significantly enhanced the aboveground biomass of maize, regardless of the induced treatments (herbivore attack or JA application) (Figure 1). A significantly lower aboveground biomass was found in N1 (52.2 mg/kg supplemented soil) maize with induced treatments, which were 78.34% and 67.53% of the biomass of untreated N1 maize, respectively (Figure 1).

### 2.2. Effects of N Supply on Herbivore Performance in Maize

The analysis of generalized linear models (GLM) showed that the relative growth rate (RGR) of *S. frugiperda* larvae was significantly affected by the induced treatments, not the N supply (Figure 2a). When maize plants grew in N2 (156.6 mg/kg of N-supplemented soil) soil, herbivore attack and JA application decreased the RGR of *S. frugiperda* larvae (Figure 2a).

N supply and the induced treatments, individually and in combination, affected the feeding leaf area (Figure 2b). N supply significantly increased the feeding leaf area of untreated maize. After the induced treatments, the feeding leaf area significantly increased in N1 maize (Figure 2b). The induced treatments significantly decreased the feeding leaf area of N2 maize compared with untreated N2 maize (Figure 2b). After JA application, the feeding leaf area of N2 maize was significantly less than that of N2 maize after herbivore attack (Figure 2b).

In the absence of induced treatments, *S. frugiperda* preferred maize plants growing in N2 soil (Figure 2c). Herbivore attack enhanced N1 maize to attract insects, while the choice of *S. frugiperda* to N2 maize was significantly decreased (Figure 2c). After JA application, the choice of N2 maize by *S. frugiperda* was also decreased (Figure 2c). Regardless of whether the maize plants were growing in N0, N1 or N2, the induced treatments enhanced the ability of the plants to repel *S. frugiperda* larvae (Figure 2d).

### 2.3. Effect of N Supply on Maize Nutrients and Secondary Metabolites

The induced treatments, individually and in combination with the N levels, affected the content of soluble protein (Figure 3a). In the absence of the induced treatments, N2 maize significantly increased the content of soluble protein (Figure 3a). Compared to untreated plants, N2 maize with JA application showed a significant reduction (Figure 3a). 

N levels, individually and in combination with induced treatments, affected the content of soluble sugar and amino acids (Figure 3b,c). In the absence of the induced treatments, the N supply significantly increased the contents of soluble sugar and amino acids (Figure 3b). However, the contents of soluble sugars in N2 maize with induced treatments were significantly lower than those of the corresponding N0 and N1 maize (Figure 3b). Compared to untreated plants, N2 maize with induced treatments had significantly reduced contents of soluble sugar (Figure 3b). N1 maize with induced treatments showed significantly increased amino acid contents. In contrast, N2 maize with induced treatments had significantly reduced amino acid contents (Figure 3c).

The induced treatments and N levels, individually and in combination, had effects on the content of total phenolics and 2, 4-dihydroxy-7-methoxy-1,4-benzoxazin-3-one (DIMBOA) (Figure 3d,e). The content of total phenolics in plants grown in N2 soil was significantly higher than that in N0 and N1 maize, regardless of herbivore attack or JA application (Figure 3d). N supply significantly increased the DIMBOA content in maize (Figure 3e). Induced treatments significantly induced DIMBOA, and significantly higher DIMBOA was detected in maize plants after herbivore attack than JA application, regardless of whether the maize grown in N0, N1 or N2 conditions (Figure 3e).

A Pearson correlation analysis showed that a positive correlation was found between the feeding leaf area and the contents of amino acids, soluble protein and soluble sugar of untreated maize (Figure 3f). Furthermore, a positive correlation was also found between the feeding leaf area and the content of amino acids of maize with the induced treatments (Figure 3f).

### 2.4. Effect of N Supply on Maize Phytohormones 

A GLM analysis showed that the induced treatments and N levels individually had effects on the level of JA and SA (Figure 4a,b). N2 maize with herbivore attack significantly enhanced the level of JA, compared with those of maize with JA application and control (Figure 4a). N2 supply increased the levels of SA, regardless of induced treatments (Figure 4b). N2 maize with both induced treatments significantly enhanced SA levels, compared with those of untreated maize (Figure 4b). 

The induced treatments and N levels, individually and in combination, affected the level of ABA. Without herbivore attack, N1 application significantly increased ABA levels (Figure 4c). More specifically, significantly higher ABA levels were detected in maize with herbivore attack than in maize with JA application (Figure 4c).

### 2.5. Effect of N Supply on Maize VOCs 

A total of 90 volatiles were collected across all maize plants growing in different N soils under different induced treatments (Appendix A). These volatiles were divided into seven categories, including aldehyde ketones, alcohols, acidic compounds, terpenes, esters, aromatics and nitrogenous compounds (Figure 5a). 

Without induced treatments, the number of identified volatiles from the maize plants was significantly increased with N-supplemented levels, and the principal components analysis (PCA) showed a clear separation between N0 maize and N1, N2 maize (Figure 5b). N2 maize emitted the most types of volatiles, where the abundance of decyl aldehyde and dodecane were significantly increased compared to N0 maize (Figure 5a,e). In particular, terpenes (e.g., 1,6-octadiene) were unique in N2 maize. Esters (e.g., indoles) from N0 maize plants were significantly higher than those from N1 maize plants but were not detected in N2 maize plants; N1 maize plants emitted the highest concentration of acrylic acid (Figure 5a,e). 

Herbivore attack significantly increased the number of volatiles emitted from maize plants growing in N soil (Figure 5a). A clear separation was found between N1 and N2 maize volatiles along the second axis (Figure 5c). β-caryophyllene and nerolidol were only identified from N1 maize plants, whereas methyl salicylate (MeSA) and jasmonates were unique in N2 maize (Figure 5f). β-laurene and phenylacetaldehyde were emitted from N1 and N2 maize, whose concentrations significantly increased with N-supplemented levels. The abundance of palmitic acid increased in N1 maize and decreased in N2 maize (Figure 5f).

After JA application, the number of volatile compounds was significantly higher, whereas esters were not identified in N1 maize (Figure 5a). A clear separation was found between N0 and N1 maize volatiles, whereas N2 maize volatiles contained N0 and N1 maize volatiles (Figure 5d). We identified β-laurene, limonene, acrylic acid, indoles and myristic aldehyde in N maize, and the abundance of acrylic acid and indoles significantly increased with N-supplemented levels in soil, whereas the abundance of limonene significantly decreased with N-supplemented levels in the soil (Figure 5g).

### 2.6. Effect of N Supply on Maize Defensive Genes 

N levels and the induced treatments, individually and in combination, affected the expression of five genes that are involved in plant defense (Figure 6). 

Under nontreated conditions, the expression of *MPI* and *PAL* was significantly upregulated in N2 maize (Figure 6b,d), while the expression of *BX9* and *LOX* was dramatically downregulated in N2 maize (Figure 6a,c); the expression of *BX9* in N1 maize was significantly lower than that in N0 maize (Figure 6a), while the expression of *LOX*, *PAL* and *AOS* was dramatically enhanced in N1 maize (Figure 6c,e). 

With herbivore attack, N1 maize showed a significantly increased expression of *MPI* and *PAL* (Figure 6b), and a reduced expression of *AOS* compared with N0 maize (Figure 6e). A significantly higher expression of *BX9* and *AOS* (Figure 6a,e) and a significantly lower expression of *MPI* and *PAL* were detected in N2 maize with herbivore attack (Figure 6b,d). A significantly higher expression of *MPI* (Figure 6b) and a significantly lower expression of *LOX* and *AOS* were detected in N1 maize with herbivore attack (Figure 6c,e). 

With JA application, N supply significantly reduced the expression of *AOS* (Figure 6e). N1 maize showed a significantly increased expression of *LOX* and a reduced expression of *BX9* (Figure 6a,c) compared with N0 maize. N2 maize showed a significantly increased expression of *PAL* compared with N0 maize (Figure 6d). A significantly higher expression of *BX9* and *MPI* was detected in N maize with herbivore attack (Figure 6a,b). Significantly higher *LOX* and significantly lower *AOS* expressions were detected in N1 maize with herbivore attack (Figure 6c,e). 

More specifically, significantly higher *MPI* expression was found in maize with JA application than in maize with herbivore attack, regardless of whether the maize plants grew in N0, N1 and N2 soil (Figure 6b); JA application also increased the expression levels of *LOX* in N1 maize, *PAL* in N2 maize, and *AOS* in N0 maize (Figure 6c,e).

## 3. Discussion

The present study demonstrated the negative effects of N supply on maize defense against *S. frugiperda* larvae, in which higher N levels increased the plant biomass (Figure 1) and enhanced nutrient contents (soluble protein, soluble sugar and amino acids) of maize plants without any treatments (herbivore attack or JA application) (Figure 3), which improved the leaf quality and, consequently, *S. frugiperda* larvae fed more maize leaves (Figure 2b). The plants growing in N increased the nutritional quality (biosynthesis or accumulation of proteins, free amino acid and sugars) that might have attracted insects [10,18,20]. Therefore, N can exert a variety of negative effects on plant defense patterns [10,12,20]. High N availability in maize significantly increased the N and total protein content; subsequently, leaf-chewing *S. exigua* larvae showed higher performance and preferentially fed on high-N maize [12]. This and our results are in accordance with the resource availability hypothesis, which predicts that plants in resource-rich environments have lower levels of defense and encounter higher levels of herbivory than plants in resource-poor environments [20,39]. Although *S.*
*frugiperda* larvae demonstrated a higher preference for N2 maize, their corresponding RGR was not increased remarkably. Probably, insects had to allocate some energies taken up from maize leaves to detoxify secondary metabolites, which was consist with the increased production of total phenolics and DIMBOA along with N levels (Figure 2). Xu et al. reported that increased N fertilization in maize enhanced the total phenolics content and ultimately decreased the RGR of *O. furnacalis* [13]. However, plant nutrition status was more positively closely associated with leaf area fed by *S. frugiperda* larvae than the production of secondary metabolites (Figure 3f).

Priming is an adaptive strategy that increases the readiness of the induced defensive capacity of plants by the stimuli from a series of biotic or abiotic factors, e.g., insect attack and chemicals [40,41]. Although priming for defense may combine the advantages of enhanced disease protection and lower costs [42], little is known about how plants set their defensive priorities to deal with many biotic and abiotic stresses that occur concurrently [43]. In the present study, *S. frugiperda* attack and JA application were considered as priming events, whereas our results show that whether priming enhanced the maize defense against *S. frugiperda* larvae depended on the N supply level. The 156.6 mg/kg of N supplementation enhanced the defense of maize plants against pests, while the 52.2 mg/kg of N supplementation increased nutritional quality without strengthening induced defenses much, making plants more attractive to the insect. Ren et al. also reported that the expression of induced defense was dependent on N availability after *S. exigua* attack, in which maize containing high levels of N significantly deterred larval feeding and had negative effects on larval performance, but maize with low levels did not affect larval performance and preference [12]. 

Nutritional quality can be induced by insect attack to generate changes [12]. Feeding-damaged maize in the 156.6 mg/kg of N treatment decreased the contents of amino acids and soluble sugars (Figure 3) and, thus, decreased larval growth and feeding leaf area (Figure 2). Insect attack also led to increased levels of phenolics in the plants supplemented with 156.6 mg/kg of N and, thus, had negative effects on insects (Figure 3). The production of phenolics in 68 woody plant studies and nine species of herbaceous plants was higher in resource-rich plants after insect damage than in resource-poor plants [44,45]. Ren et al. also reported similar results [12]. JA is a key signaling compound that triggers induced responses in plants [46]. Exogenous JA application enhances the defense response of crops in response to biotic stress in physiological and biochemical states [47,48,49]. In the present study, JA application decreased the soluble sugar content and increased the total phenolics in maize supplemented with 156.6 mg/kg of N to reduce insect growth and the feeding area by *S. frugiperda* larvae (Figure 2 and Figure 3), which was consistent with the results of Rasmann et al., who reported that methyl JA (MeJA) priming in *Arabidopsis thaliana* and tomato *S. lycopersicum* led to the reduced weight of *Pieris rapae* caterpillars [50]. Qi et al., also showed that MeJA treatment reduced the performance of Asian corn borer *O.*
*furnacalis* in maize [36]. In contrast, feeding damage or JA applied to maize in the maize supplemented with 52.2 mg/kg of N increased the contents of amino acids and did not significantly change the levels of total phenolics (Figure 3). Moreover, the feeding preference of *S. frugiperda* was found to be significantly higher in the maize supplemented with 52.2 mg/kg of N (Figure 2). This result is consistent with the “plant stress hypothesis”, which assumes that stressed plants (with 52.2 mg/kg of N considered as N deficiency) serve as a more suitable host for herbivorous insects due to increased nutritional value, i.e., amino acids, and the reduced syntheses of plant defensive compounds [51]. Therefore, plants in resource-rich environments have a higher plasticity of induced defense to depress further insect attacks than in resource-poor environments.

N is an abiotic factor that alters the release of herbivore-induced VOCs in plants [10,30]. Gouinguené and Turlings reported that unfertilized maize plants release significantly lower amounts of herbivore-induced VOCs than fertilized plants [29]. However, Schmelz et al. reported a negative relationship between N availability and volicitin-induced VOC emissions in maize [32]. Our results show that the number and quantity of VOCs from maize plants significantly increased with N levels (Figure 5). The content of volatiles was directly proportional to the amount of fertilization, with the exception of β-laurene, α-citrus oleene and (E)-β-farnesene [4]. Indole, a volatile aromatic compound, has a direct defense against pests and was only detected in N0 and N1 maize (Figure 5). Veyrat et al. reported that indole decreases the food consumption, plant damage and survival of *S. littoralis* caterpillars, which consistently avoid indole-producing plants in olfactometer experiments and feeding assays [52]. In addition, the enhanced emissions of terpenoids lead to a stronger attraction of natural enemies [4], and terpene volatiles also influence the behavioral response of *Bemisia tabaci* to tomato [53]. These results were well corroborated by the finding that *S. frugiperda* larvae demonstrated a better performance in N2 maize. 

Herbivore attacks trigger the emissions of VOCs that have been well studied [4]. *S. frugiperda* larval attack significantly increased the amounts of VOCs, including aromatics, aldehyde ketones, alcohols and terpenes (Figure 5). After insect attack, both rice and maize can release MeSA and MeJA, which are key signaling compounds involved in plant-induced responses [36]. The application of MeSA to maize strongly repels leafhoppers *Cicadulina storeyi* [54], which is consistent with the less frequent choice of *S. frugiperda* larvae for maize plants supplemented with 156.6 mg/kg of N (MeSA and jasmonates were only identified in 156.6 mg/kg of N treated maize). The volatile β-caryophyllene, a common volatile in plants, has a double effect on insects, as it can repel a number of agriculturally important pests, including *Tribolium castaneum* and *Liposcelis bostrychophila*, and attract *Apis cerana*, *Helicoverpa assulta* and *Spodoptera litura* [55,56,57,58]. Gao et al. and Liu et al. showed that low concentrations of β-caryophyllene had an attractive effect on *Bactrocera dorsalis* and *Drosophila suzukii* females [59,60]. Our results also indicate that a low concentration of β-caryophyllene and nerolidol was only emitted from maize supplemented with 52.2 mg/kg of N (Figure 5), which was probably why *S. frugiperda* larvae preferred 52.2 mg/kg of N maize with herbivore attack. Lou & Baldwin reported that N supply did not influence MeJA-induced VOCs [21]. JA application did not increase VOC emissions in the plants supplemented with 156.6 mg/kg of N (Figure 5). However, significantly higher volatiles of acrylic acid, indoles and myristic aldehyde (Figure 5) could explain the reduced selection of maize supplemented with 156.6 mg/kg of N by *S. frugiperda* larvae (Figure 2).

Numerous studies have shown that exogenous JA application can induce maize defense [42,61,62,63,64]. In this study, JA application enhanced the insect resistance of maize growing in 156.6 mg/kg of N more intensely, mainly reflecting in a smaller feeding leaf area than that of maize with herbivore attack. However, the levels of anti-herbivore signal substances (i.e., JA and ABA) and DIMBOA were significantly lower than those of herbivore-attacked maize (Figure 3 and Figure 4). The expression of anti-herbivore defensive genes from plants with exogenous JA application, including *MPI* and *PAL*, was significantly higher than that of herbivore-attacking plants (Figure 6). *MPI* encodes a maize proteinase inhibitor (MPI) protein that is able to inhibit the two types of insect digestive proteinases, elastases and chymotrypsin [65,66]. Paulillo et al. reported that soybean proteinase inhibitors affected trypsin and chymotrypsin activities and the development of *S. frugiperda* larvae [67]. *PAL* encodes phenylalanine ammonia lyase (PAL), which catalyzes the deamination of phenylalanine to cinnamic acid, the entry and key regulatory step into the phenylpropanoid pathway [67]; the products of the stress-induced phenylpropanoid pathway are commonly lignin or lignin-like polymers [68,69]. The high content of lignin could make plants less palatable to herbivores [11]. Thus, the expression of two defensive genes could explain the stronger defense found in the maize supplemented with 156.6 mg/kg of N after JA application. A significantly higher content of indoles (considered direct defense substances) may contribute to decreased food consumption by *S. frugiperda* larvae and plant damage (Figure 5g).

## 4. Materials and Methods

### 4.1. Soils, Plants and Insects

The experimental soil was collected from the top layer (5–25 cm) of the conventional maize field at the Agriculture Experiment Station (23°08′ N, 113°15′ E) of South China Agriculture University (SCAU), Guangzhou, China. The soil was a red clay loam with a pH of 6.38, containing 15.98 g/kg of organic matter, 1.089 g/kg of total nitrogen (N), 0.959 g/kg of total phosphorus (P), 24.26 g/kg of total potassium (K), 60.63 mg/kg of available N, 173.89 mg/kg of available P, and 370.73 mg/kg of available K. A certain amount of prepared urea solution for the required concentrations was added into plastic pots (10 cm height, 9 cm diameter) with 500 g of soil. The soil was irrigated with distilled water to maintain the soil moisture at approximately 60–70% of the water holding capacity by weight.

Seeds of the maize *Z. mays* Nongtian 88 were provided by the Department of Breeding, College of Agriculture, SCAU. The fall armyworm *S. frugiperda* (Lepidoptera: Noctuidae) was hatched from eggs by the Insectarium of the Institute of Tropical and Subtropical Ecology, SCAU. Upon hatching, the 1st instar larvae were reared on artificial diets, as described previously [70]. The rearing was carried out under constant conditions of 27 ℃, 65% relative humidity and a 12 h dark/12 h light cycle.

### 4.2. Experimental Design and Conditions

We conducted a 2 × 2 full factorial experiment, in which we manipulated N-supplemented soil, *S. frugiperda* attack and jasmonic acid (JA) application as the two main factors. Three levels of N were added to the soil for the growing maize plants, namely, (1) no N addition (N0); (2) the addition of 52.2 mg/kg of N (N1); (3) the addition of 156.6 mg/kg of N (N2). The second factor was used: (1) untreated (control, CK); (2) *S. frugiperda* attack (SF); (3) jasmonic acid application (JA). In total, we constructed 9 treatment combinations and more than 4 replicates for each treatment combination in every measurement.

Maize seeds were surface-sterilized with 5% NaClO_3_ for 5 min and then rinsed extensively in distilled water before germination on moist filter paper for 48 h at 25 ℃ in the dark. Same-size maize seedlings were then individually transplanted into plastic pots (10 cm height, 9 cm diameter), filled with soil containing different urea concentrations, and randomly placed in a glasshouse at 25 ± 5 ℃ with a photoperiod of 16 h light/8 h dark.

When the maize seedlings grew to the six-leaf stage, three third-instar larvae of *S. frugiperda* previously starved for 12 h were applied to each plant on one-third of the maize plants. Each plant, including the control plants, was then covered with a fine-mesh bag (10 μm) to prevent the *S. frugiperda* caterpillars from escaping. Finally, at 12 h after herbivore inoculation, damage to the maize leaves was observable. In addition, 20 μL of 4 mmol/L JA (Sigma) per plant was brushed on the last second leaf of one-third of the maize plants.

### 4.3. Plant Aboveground Biomass

After herbivore attack or JA application for 12 h, the maize plants were allowed to grow for 6 days. Then, the maize plants growing in soil supplemented with different N levels were sampled, and 6 randomly plants were selected from every treatment combination. We cut the aboveground parts of maize plants to determine their fresh weights and then vacuum dried them at 60 ℃ for 72 h to obtain biomass.

### 4.4. Herbivore Performance

#### 4.4.1. Growth of *S. frugiperda*

After herbivore attack or JA application for 12 h, we fed 3rd instar larvae with fresh leaves of maize growing in soil supplemented with different N levels. After measuring the initial weight of the larvae, 10 larvae from every treatment combination were transferred to plastic boxes (4 cm diameter) to be reared individually with enough fresh leaves daily; each bioassay was performed in triplicate. We measured the weight of larvae after 7 d. The relative growth rate (RGR) was calculated as RGR (%) = (W_n_ − W_0_) / (W_0_ × T) × 100% [71], where W_0_ was the initial weight of *S. frugiperda*, W_n_ was the fresh weight at the check day, and T (d) was the duration of the experimental period.

#### 4.4.2. Feeding Behavior of *S. frugiperda*

We studied the preference of *S. frugiperda* larvae for the leaves of maize growing in soil supplemented with different N levels. The 3rd fully expanded leaves counted from the top of each plant were sampled. Leaf discs from different treatments were cut with a leaf puncher (40 mm × 40 mm) and placed in a Petri dish (15 mm diameter). Twenty healthy 3rd instar larvae previously subjected to starvation for 12 h were used for every treatment combination and were placed in the middle of each Petri dish individually for 5 h. To quantify damage, the remaining leaf pieces were scanned by a scanner, and the feeding leaf area was calculated by using Adobe Photoshop CS6 software (Adobe Systems, San Jose, CA, USA).

#### 4.4.3. *S. frugiperda* Preference as Assessed by Olfactometer

The behavioral responses of *S. frugiperda* larvae to plants growing in soil supplemented with different N levels were measured in a four-arm olfactometer (Appendix A). The olfactometer consisted of an internal star-shaped exposure arena [5-cm internal diameter (ID), 10-cm height] with four arms (1.5 cm ID, 5.5 cm length), which were held together with plastic nuts and bolts. The areas narrowed towards the perimeter and were connected to glass chambers holding the corresponding plants (Appendix A) with Teflon tubing, or to glass arms through a 3 mm diameter hole at the end of each of the four arms. Prior to each experiment, all glassware was washed with Teepol detergent, rinsed with acetone and distilled water and baked in an oven overnight at 130 ℃. Perspex components were washed with Teepol solution, rinsed with 80% ethanol solution and distilled water and left to air dry.

One *S. frugiperda* larva previously starved for 12 h was carefully introduced into the central chamber of the olfactometer with a small brush. Air was drawn through the central hole by a vacuum pump (220–240 VAC; Charles Austen Pumps Ltd., Byfleet, Surrey, UK) and thereby pulled through each of the four side arms (75 mL/min/arm) and subsequently exhausted from the room. Each larva was allowed to acclimatize for 2 min, after which the experiment was run for 10 min, and the olfactometer was rotated by 90 deg every 5 min to control for any directional bias. The larvae that climbed to more than one-third of the end of an arm and stayed for more than 30 s were considered to have made their choice of the plant arm; individuals who did not choose within 5 min were excluded from the analyses. A total of 30 larvae were individually assayed for each treatment combination. All tests were conducted between 10:00 and 16:00 in a climate-controlled laboratory room (27 ± 3 ℃, 40% RH).

After replacing different plants, the interior of the olfactometer and the taste source bottle were cleaned with 60–90% ethanol and dried with a hair dryer. After each treatment test, the Teflon tube, washing cylinder and four-arm olfactory tubes were soaked in 30% ethanol solution for cleaning, spray washed with 90% ethanol and dried naturally to eliminate the odor influence between different the treatments.

### 4.5. Measurement of Total Protein, Soluble Sugar and Amino Acids

Total protein was extracted using the trichloroacetic acid-acetone method according to the procedures of Damerval et al. and Jellouli et al. [72,73]. The protein concentration was determined according to Bradford’s method using bovine serum albumin as a standard [74].

The enthrone-sulfuric acid colorimetric assay was used to determine the content of soluble sugar [75]. Enthrone was dissolved in 80% sulfuric acid to prepare the enthrone-sulfuric acid reagent before use. The calibration curve was constructed based on the various glucose concentrations and their corresponding absorbance was determined by a spectrophotometer (Shimadzu, Kyoto, Japan) at 620 nm.

The concentrations of total amino acids were measured with kits purchased from the Nanjing Jiancheng Biological Engineering Institute (NJBI, Nanjing, China). Briefly, 100 mg of fresh leaves from each sample was homogenized with 900 μL of distilled water with a homogenizer (Fast Prep (FP120), Thermo, Savant, Hyannis, MA, USA). After centrifugation at 3500 rpm for 10 min, the resulting supernatants were measured with a plate reader at 650 nm (Molecular Devices, San Jose, CA, USA). The amino acid content was calculated on the basis of a calibration curve by the glycine provided by the kit.

### 4.6. Secondary Metabolites Analyses

To determine whether N supply alters the main secondary metabolites in maize, the concentrations of total phenolics and DIMBOA were measured, with 6 replications per treatment combination. The total phenolic concentration was determined using colorimetric kits purchased from the Nanjing Jiancheng Bioengineering Institute (NJBI, Nanjing, China). Briefly, 100 mg of leaf material from each sample with 900 μL of ice-cold 60% (*v*/*v*) ethanol was shaken and mixed in an ice-bath, and then centrifugated at 4 °C and 12000 rpm for 10 min. Finally, 200 μL of the resulting supernatant from each reaction was transferred to a clear 96-well microplate, and the absorbance was measured at 760 nm with a Multifunctional Microplate Reader (Molecular Devices, San Jose, CA, USA). A standard curve was prepared with gallic acid concentrations ranging from 0 to 10 mg/mL, and the results were expressed as the gallic acid equivalent (GAE mg/g dry leaf).

Approximately 100 mg of leaf tissues was extracted with 1 mL of the extraction solution (50% methanol containing 0.5% formic acid and 10 μg of 4-methylumbelliferone as the internal standard). The concentrations of DIMBOA were analyzed on an HPLC–MS/MS system (LCMS-8040, Shimadzu, Japan) following the method described by Gao et al. [76].

### 4.7. Measurement of Phytohormones

The concentrations of phytohormones, including JA, salicylic acid (SA) and abscisic acid (ABA) were measured using the corresponding kits obtained from the Nanjing Jiancheng Biological Engineering Institute (NJBI, Nanjing, China), with 4 replications for each treatment combination. Leaf samples (0.15 g) from each sample were ground into a homogenized powder in liquid nitrogen and transferred into a 2 mL centrifuge tube with 1.5 mL 0.1 mol/L of phosphate buffered saline (PBS) (pH 7.2). After centrifugation (6000 rpm, 15 min, 4 °C), 10 μL of the resulting supernatant was loaded into the well of the ELISA plate to which 40 μL of PBS had been previously added. The ELISA plate was wrapped with aluminium foil and then incubated at 37 °C for 30 min. The plate was washed five times with PBS. Then, 100 μL of chromogenic solution was added to each well, and the plate was incubated at 37 °C for 15 min in the dark. Finally, 50 μL of termination solution was added to each well. The absorbance of phytohormones was measured at 450 nm with a microplate reader (Molecular Devices, San Jose, CA, USA). The concentration of phytohormones was calculated using a five-point standard curve (JA: 62.5, 125, 250, 500 and 1000 pmol/L; SA: 75, 150, 300, 600, 1200 pmol/L; ABA: 18.75, 37.5, 75, 150 and 300 μg/L) that was developed with the corresponding standards (supplied with the kits). The test results were validated with both positive and negative controls.

### 4.8. Volatile Organic Compound (VOC) Collection and Quantification

To determine the number and quantity of volatiles emitted from the above plants, headspace volatile samples were collected in a climate room (25 ± 2 ℃, RH 60–65%). In brief, two hours before volatile trapping, the soil in pots was carefully wrapped with aluminum foil. One plant (four plants per each treatment combination) was placed in a 1-L glass jar. The jars were closed with a Viton-lined inert glass lid with an inlet and an outlet. Air was removed using a vacuum pump at 500 μL/min, and the incoming air was purified through a stainless-steel cartridge filled with 200 mg Tenax-TA (20/35-mesh; Grace-Alltech, Deerfield, MA, USA). A similar cartridge was used to trap the emitted plant volatiles at the outlet. After 10 h of trapping under continuous light (4750 ± 86 lx), the trap was rinsed with 250 μL of N-hexane. For each treatment combination, the volatile collection was repeated five to seven times, and collections were performed for each treatment in parallel on each experimental day in replicates.

Samples were analyzed with a GC–MS–2010 instrument (Shimadzu Corp., Kyoto, Japan) consisting of an Agilent 6890 Series gas chromatograph (Agilent, Santa Clara, CA, USA) that was coupled to a 2010 Plus single quadrupole mass spectrometer. Separations were performed using an HP-5 capillary column (JB DB-WAX, 29 m, 0.25 mm id, 0.25 µm film thickness, Agilent Technologies, Santa Clara, CA, USA) with helium as the carrier gas. The initial oven temperature of 50 °C was maintained for 3 min and then raised to 230 °C at a rate of 8 °C/min, with an additional 7.5 min of hold time. The gas flow was set to achieve a constant linear velocity of 2 μL/min, with no split. The total run time was 50 min, and the injection temperature was set at 250 °C. The mass spectrometer was operated in electron impact (70 eV) in the interval of 50–550 *m*/*z* with a scan velocity of 5000 amu/s and a solvent cut time of 1.9 min. Before the analysis, a solvent blank was analyzed (pure n-hexane solution), and then an analytical blank was obtained by extracting an unexposed cartridge with 2 mL of n-hexane solution. Peak areas were automatically normalized to the standard internal area, and metabolite identification was performed by comparing each peak’s mass spectrum with the NIST library collection (NIST, Gaithersburg, MD, USA). Relative quantification was based on the peak area of each component of the volatiles. The linear index difference max tolerance was set at 10, while the minimum matching for the NIST library search was set at 85%. Using the Kovats index, identification was further confirmed by analyzing a series of hydrocarbons (C10-C26).

### 4.9. Quantitative Real-Time PCR Analysis of Maize Defensive Genes

Total RNA was extracted using TRIzol reagent (Invitrogen, Carlsbad, CA, USA) and treated with RNase-free DNase I to remove DNA contamination. The RNA purity and integrity were checked by ensuring that absorbance ratios (A260/280) were between 1.8 and 2.0 and by agarose gel (1.5%) electrophoresis. Gene-specific primers for phenylalanine ammonia-lyase (*PAL*) (GenBank Accession no. L77912) (5′-AGA AGG TGA ACG AGC TGG A-3′ and 5′-TTG TCG TTC ACG GAG TTG A-3′); maize proteinase inhibitor (*MPI*) (X78988) (5′-ACA ACC AGC AGT GCA ACA AG-3′ and 5′- GAA GAT GCG GAC ACG GTT AG-3′); allene oxide synthase (*AOS*) (AY488135.1) (5′-ACC GGT GTC ACG AAA GCT AC-3′ and 5′-AGC GAC AAA CAC CTC CAA TC-3′); lipoxygenase (*LOX*) (DQ335760.1) (5′-TGT ACG TGC CGA GGG ACG AG-3′ and 5′-CGA GCG TCT CCC TCG CGA ACT C-3′); and glucosyltransferase (*BX9*) (AF331855) (5′-GCA ACA TGA GGT ACG TGT GC-3′ and 5′-GCA GCG ATC TTG AAT TCC TT-3′) were designed. The primers for the housekeeping gene ubiquitin (*UBQ*) (NM_001138130), which was used as the endogenous control, were UBQS (5′-GCA GTG CTG CAG TTC TAC AAG G-3′) and UBQR (5′-GCA GTA GTG GCG GTC GAA GTG G-3′).

A qPCR was performed on a DNA Engine Opticon 2 Continuous Fluorescence Detection System (MJ Research Inc., Waltham, MA, USA) with a SYBR Premix Ex Taq Kit (Takara, Japan) under the thermal program of one cycle of 95 ℃ for 10 s, followed by 40 cycles of 95 °C for 5 s and 60 °C for 30 s. After the qRT-PCR, the homogeneity of the PCR product was confirmed by a melting curve analysis. The relative copy number of the mRNAs of the genes outlined above was calculated according to the 2^–∆∆CT^ method [77]. The threshold cycle value difference (^∆^CT) between the mRNA of the assessed genes and UBQ RNA in each reaction was used to normalize the level of total RNA.

### 4.10. Statistical Analysis

All statistical analyses were conducted using the software package SPSS (version 22; SPSS, Inc., Chicago, IL, USA). For all tests, a significance level of 5% was used. Descriptive statistics followed by exploration were used to test the data for normality, and all of them fulfilled the assumption of normal distribution and homogeneity of variance. Univariate GLM distinguished the effects of N addition, treatments and their interactions on the measured parameters. The correlations between feeding leaf area and the contents of nutrients and secondary metabolites were analyzed by the Pearson correlation coefficient. The data were log-transformed when necessary to comply with variance homogeneity.

## 5. Conclusions

In the absence of herbivore attack or exogenous JA application, the N supply demonstrated negative effects on maize defense against *S. frugiperda* larvae, with higher N levels increasing plant biomass and enhancing the nutrient (soluble protein, soluble sugar and amino acids) contents of maize plants to improve leaf quality. Consequently, *S. frugiperda* larvae fed more maize leaves. When these results were considered together with the enhanced secondary metabolites (total phenolics) and phytohormones (SA) in the maize supplemented with 156.6 mg/kg of N, we assumed that maize nutrient status, not secondary metabolites, drove herbivory without any induced treatments. Herbivore attack or JA application induced maize defense in an N-dependent manner (Figure 7). After herbivore attack, the maize supplemented with 52.2 mg/kg of N maize increased the content of amino acids, not total phenols; significantly downregulated the expression of *AOS* related to JA synthesis; and emitted β-caryophyllene and nerolidol at low concentrations to attract pests. Conversely, the maize supplemented with 156.6 mg/kg of N decreased the contents of soluble sugar and amino acids, and increased defensive chemicals, e.g., total phenols, SA and ABA, to reduce the palatability to *S. frugiperda* larvae; the expression of *AOS* significantly upregulated, and β-laurene, methyl salicylate, jasmonates and other volatiles were emitted to repel pests. Compared with herbivore attack, JA application enhanced insect resistance in maize supplemented with 156.6 mg/kg of N more intensely, mainly reflecting a smaller feeding leaf area, which might be attributed to the two upregulated defensive genes, *MPI* and *PAL*. Therefore, the optimal N level (156.6 mg/kg of N) and appropriate priming (e.g., JA application) not only enhanced plant biomass, but also induced a relatively high defense chemical content and abundant volatiles to defend against pests. Thus, more research will be needed to investigate the effects of N supply or other environmental factors on plant-insect interactions, which will help to conduct rational fertilization and pest control in the agroecosystem.

## Figures and Tables

**Figure 1 ijms-23-10457-f001:**
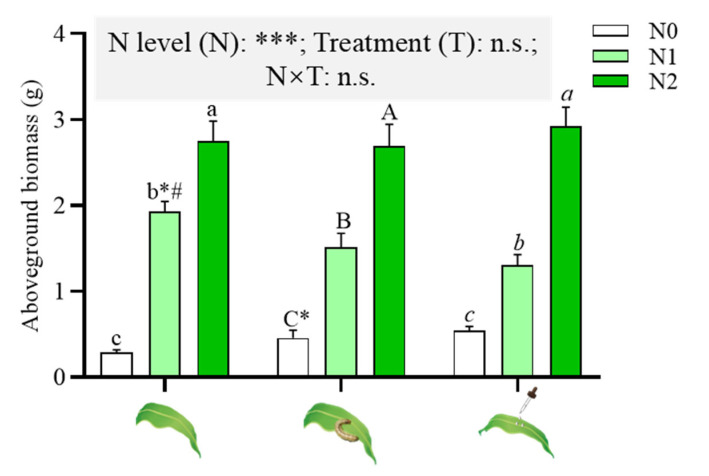
Effects of Nitrogen (N) supply on the aboveground biomass of maize under different induced treatments (herbivore attack or JA application). Generalized linear models (GLM) results of testing for N level and induced treatment effects are shown (n.s., not significant; * *p* < 0.05; *** *p* < 0.001); different lowercase, uppercase and italic lowercase letters above the bars indicate significant differences among different N levels (supplemented with 0, 52.2, 156.6 mg/kg of N, denoted as N0, N1, N2) under the same induced treatments at *p* < 0.05 (one-way analysis of variance (ANOVA)). The asterisk “*” above the bars indicates significant differences between maize with and without *S. frugiperda* attack; “#” above the bars indicates significant differences between maize with and without JA application by *t*-test.

**Figure 2 ijms-23-10457-f002:**
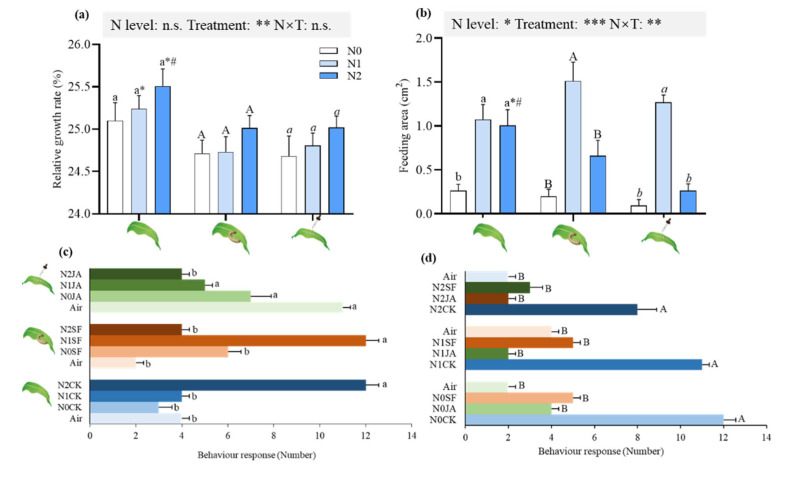
Effects of N supply on preference of *S. frugiperda* larvae towards maize under different induced treatments. (**a**) Relative growth rate (RGR) of *S. frugiperda* larvae; (**b**) feeding leaf area (cm^2^) by *S. frugiperda* larvae; (**c**) behavioral response (number) of *S. frugiperda* towards maize plants from different N levels under the same induced treatments; (**d**) behavioral response (number) of *S. frugiperda* towards maize plants from different induced treatments under the same N levels. GLM results of testing for N level and induced treatment effects are shown (n.s., not significant; * *p* < 0.05; ** *p* < 0.01; *** *p* < 0.001); different lowercase, uppercase and italic lowercase letters above the bars indicate significant differences among different N levels under the same induced treatments. The asterisk “*” above the bars indicates significant differences between maize with and without *S. frugiperda* attack; “#” above the bars indicates significant differences between maize with and without JA application.

**Figure 3 ijms-23-10457-f003:**
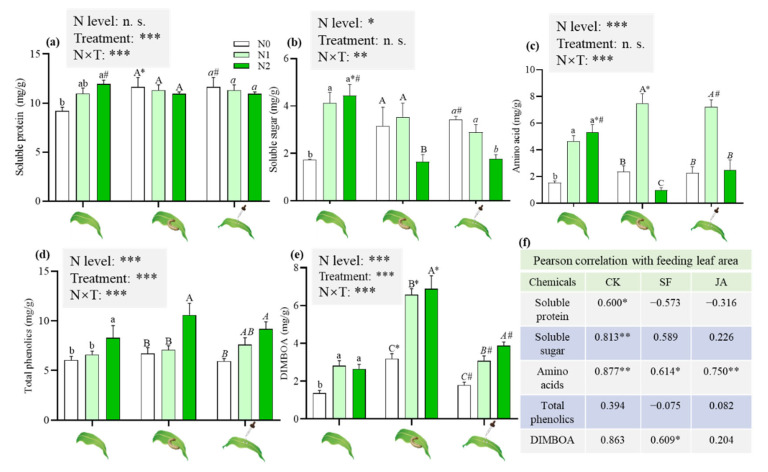
Effects of N supply on the nutrient and secondary metabolites of maize under different induced treatments. GLM results of testing for N level and induced treatment effects are shown (n.s., not significant; * *p* < 0.05; ** *p* < 0.01; *** *p* < 0.001); different lowercase, uppercase and italic lowercase letters above the bars indicate significant differences among different N levels under the same induced treatments. The asterisk “*” above the bars indicates significant differences between maize with and without *S. frugiperda* attack; “#” above the bars indicates significant differences between maize with and without JA application. Pearson correlation analysis results are shown (* *p* < 0.05; ** *p* < 0.01).

**Figure 4 ijms-23-10457-f004:**
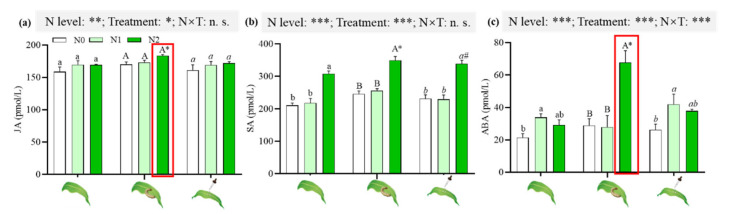
Effects of N supply on the phytohormones of maize under different induced treatments. The red box shows the significantly higher phytohormones in *S. frugiperda* attacked maize than in untreated and JA treated maize when plants were grown in N2 soil. GLM results of testing for N level and induced treatment effects are shown (n.s., not significant; * *p* < 0.05; ** *p* < 0.01; *** *p* < 0.001); different lowercase, uppercase and italic lowercase letters above the bars indicate significant differences among different N levels under the same induced treatments. The asterisk “*” above the bars indicates significant differences between maize with and without *S. frugiperda* attack; “#” above the bars indicates significant differences between maize with and without JA application.

**Figure 5 ijms-23-10457-f005:**
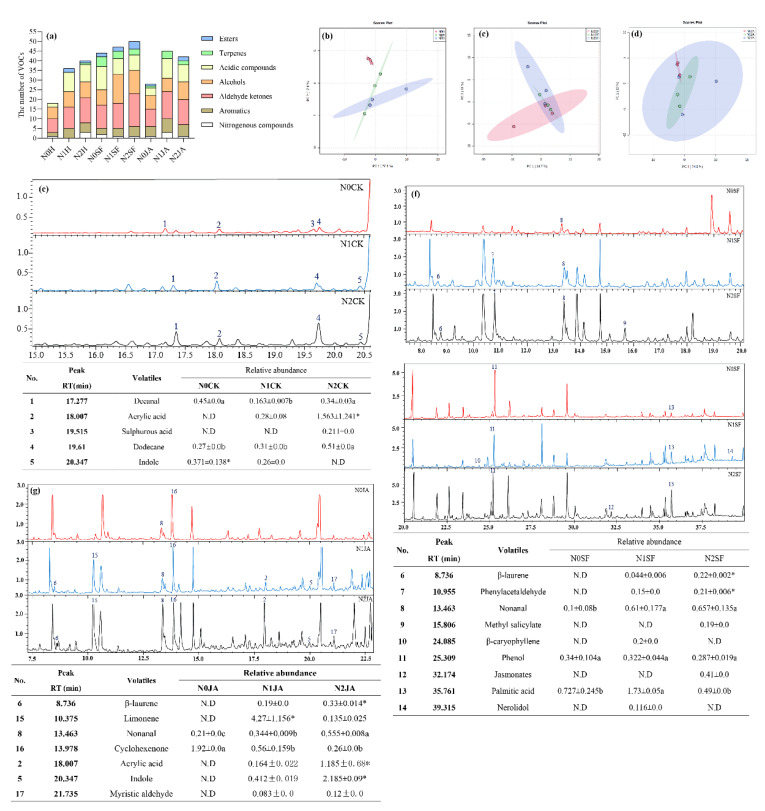
Effects of N supply on volatile organic compounds (VOCs) of maize under different induced treatments. (**a**) The number of volatiles emitted from maize plants from different treatment combinations; (**b**–**d**) the principal components analysis (PCA) of volatiles emitted from maize plants that were untreated, attacked by *S. frugiperda* or treated with JA, respectively. (**e**–**g**) GS-MS chart and the content of the main volatiles emitted by maize plants that were untreated, attacked by *S. frugiperda* or treated with JA, respectively. RT, retention time. Values followed by the same lowercase letters in the same row were not significantly different among the three N levels. Values followed by the asterisk “*” indicate significant differences between two N levels. “N.D” in the tables indicates not detected.

**Figure 6 ijms-23-10457-f006:**
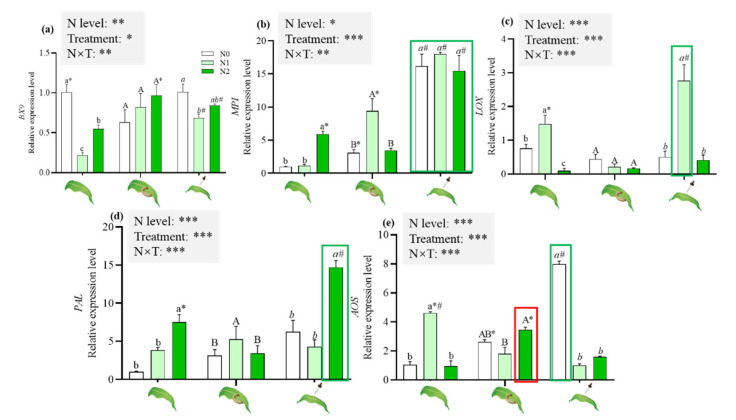
Effects of N supply on the genes involved in the induced defense of maize under different induced treatments. The red box shows a significantly higher *AOS* expression in *S. frugiperda* attacked maize than in JA treated maize. The green box shows a significantly higher gene expression in JA treated maize than in *S. frugiperda* attacked maize. GLM results of testing for N level and induced treatment effects are shown (n.s., not significant; * *p* < 0.05; ** *p* < 0.01; *** *p* < 0.001); different lowercase, uppercase and italic lowercase letters above the bars indicate significant differences among different N levels under the same induced treatments. The asterisk “*” above the bars indicates significant differences between maize with and without *S. frugiperda* attack; “#” above the bars indicates significant differences between maize with and without JA application. Abbreviations in the figure indicate the following: *BX9* (glucosyltransferase); *MPI* (maize proteinase inhibitor); *LOX* (lipoxygenase); *PAL* (phenylalanine ammonia-lyase); and *AOS* (allene oxide synthase). The same abbreviations apply below.

**Figure 7 ijms-23-10457-f007:**
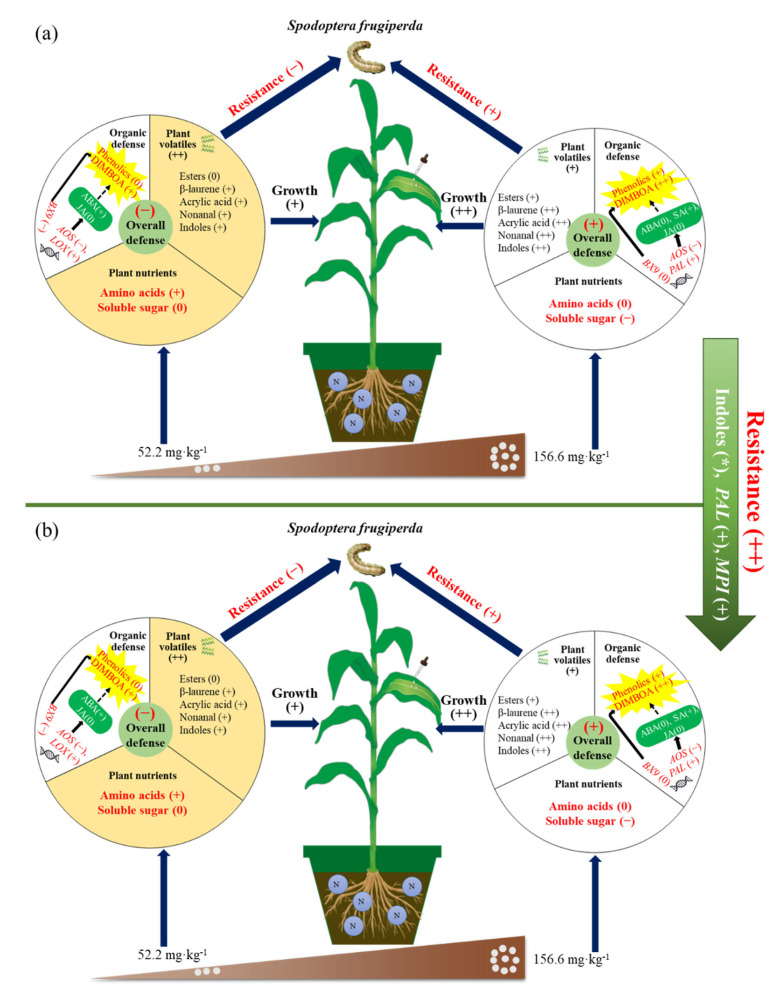
Mechanisms of N supply affecting the induced defense against *S. frugiperda* larvae. (**a**) For herbivore attack treatment; (**b**) for JA application treatment. The plus sign (+) indicates the enhanced insect resistance of maize, the increased content of chemicals (volatiles, metabolites and phytohormones), and the upregulated expression of genes. Two (++) plus signs indicate stronger effects. The minus sign (–) indicates the weakened insect resistance of maize, the reduced content of chemicals (volatiles, metabolites and phytohormones), and the downregulated expression of genes. “0” indicates no changed measured parameters. “*” indicates that the volatiles that are unique to maize plants.

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
