# Peer review of "Effects of Nitrogen Supply on Induced Defense in Maize (Zea mays) against Fall Armyworm (Spodoptera frugiperda)"

_ijms, 2022, doi:10.3390/ijms231810457_

Round 1

Reviewer 1 Report

This manuscript reports a complex and comprehensive study of the effect of nitrogen fertilization in which, in addition to insect performance, there is a thorough analysis of plant response. I request only minor revisions that address the following points:

  • An initial list of abbreviations would help in reading the text.

  • Figures 1, 3, 4, 6: Correspondence between bar colors and nitrogen levels is missing in the legend (although it can be guessed, it should be included).

  • What is the difference between figures 2e and 2f? Are these two experiments? Only one experiment with the olfactometer is described in Materials and Methods.

  • (lines 273-277) The following is stated in the discussion: “The present study demonstrated the negative bottom-up effects of N supply on maize defense against S. frugiperda larvae, in which higher N levels increased plant biomass (Fig. 1) and enhanced nutrient contents (soluble protein, soluble sugar and amino acids) of maize plants without any treatments (herbivore attack or JA application) (Fig. 3), which improved the leaf quality as a food resource for S. frugiperda larvae (Fig. 2)”. In Figure 2 we observe a trend of increasing Relative Growth Rate with increasing nitrogen supply, although the difference is not significant. However, the leaf area consumed is proportionally much higher in the case of plants supplied with nitrogen. How is it that if the leaves are more nutrient-rich and more of it is consumed, there is only a modest increase in growth rate, which is not significant compared to the control? I would like, if possible, for the authors to better discuss this finding.

Author Response

Response to Reviewer 1 Comments

Point 1: An initial list of abbreviations would help in reading the text.

Response 1: Dear Reviewer, We greatly appreciate your patient and valuable comments. We have thoroughly considered all the comments and improved our manuscript accordingly. We have made a initial list of abbreviations in the revision.

Abbreviations

The following abbreviations are used in this manuscript:

N

nitrogen

P

phosphorus

K

potassium

VOCs

volatile organic compounds

MPI

maize proteinase inhibitor

PAL

phenylalanine ammonia-lyase

BX9

glucosyltransferase

LOX

lipoxygenase

AOS

allene oxide synthase

UBQ

ubiquitin

ABA

abscisic acid

SA

salicylic acid

JA

jasmonic acid

MeJA

methyl jasmonate

MeSA

methyl salicylate

TCA

tricarboxylic acid

RGR

relative growth rate

DIMBOA

2, 4-dihydroxy-7-methoxy-1,4-benzoxazin-3-one

GC-MS

gas chromatography-mass spectrometer

PBS

phosphate buffered saline

Point 2: Figures 1, 3, 4, 6: Correspondence between bar colors and nitrogen levels is missing in the legend (although it can be guessed, it should be included).

Response 2: We revised the Figures 1, 3, 4, 6 as the followings.

Figure 1 (in the attachment)

Figure 3 (in the attachment)

Figure 4 (in the attachment)

Figure 6 (in the attachment)

Point 3: What is the difference between figures 2e and 2f? Are these two experiments? Only one experiment with the olfactometer is described in Materials and Methods.

Response 3: Figure 2e means behavioral response (number) of S. frugiperda towards maize plants from treatments of different N levels under untreated, S. frugiperda larvae attack or jasmonic acid application. Figure 2f means behavioral response (number) of S. frugiperda towards maize plants from treatments of same N levels under untreated, S. frugiperda larvae attack or jasmonic acid application.

In the revision, we revised the figure 2 (Figure in the attachment) as the following:

Figure 2. Effects of N supply on preference of S. frugiperda larvae towards maize under different induced treatments. (a), Relative growth rate (RGR) of S. frugiperda larvae; (b), feeding leaf area (cm2) by S. frugiperda larvae; (c), behavioral response (number) of S. frugiperda towards maize plants from different N levels under the same induced treatments; (d), behavioral response (number) of S. frugiperda towards maize plants from different induced treatments under the same N levels.

These are two experiments. In Materials and Methods, the four arms olfactometer were connected to glass chambers, holding the corresponding plants (Supplementary Table S1). We have revised the Supplementary Table S1 as the following:

Supplementary Table S1 The olfactometer bioassays conducted in this study

Bioassay

Arm 1

Arm 2

Arm 3

Arm 4

1

N0H

N1H

N2H

air

2

N1SF

N1SF

N1SF

air

3

N2SF

N2SF

N2SF

air

4

N0H

N0SF

N0JA

air

5

N1H

N1SF

N1JA

air

6

N2H

N2SF

N2JA

air

7

N0H

N1H

N2H

air

8

N0SF

N1SF

N2SF

air

9

N0JA

N1JA

N2JA

air

Point 4: (lines 273-277) The following is stated in the discussion: “The present study demonstrated the negative bottom-up effects of N supply on maize defense against S. frugiperda larvae, in which higher N levels increased plant biomass (Fig. 1) and enhanced nutrient contents (soluble protein, soluble sugar and amino acids) of maize plants without any treatments (herbivore attack or JA application) (Fig. 3), which improved the leaf quality as a food resource for S. frugiperda larvae (Fig. 2)”.

In Figure 2 we observe a trend of increasing Relative Growth Rate with increasing nitrogen supply, although the difference is not significant. However, the leaf area consumed is proportionally much higher in the case of plants supplied with nitrogen. How is it that if the leaves are more nutrient-rich and more of it is consumed, there is only a modest increase in growth rate, which is not significant compared to the control? I would like, if possible, for the authors to better discuss this finding.

Response 4: Thanks for your comments. We revised this paragraph as “The present study demonstrated the negative effects of N supply on maize defense against S. frugiperda larvae, in which higher N levels increased plant biomass (Fig. 1) and enhanced nutrient contents (soluble protein, soluble sugar and amino acids) of maize plants without any treatments (herbivore attack or JA application) (Fig. 3), which im-proved the leaf quality and consequently S. frugiperda larvae fed more maize leaves (Fig. 2b). The plants growing in N increased the nutritional quality (biosynthesis or accumula-tion of proteins, free amino acid and sugars) that might have attracted insects [10, 18, 20]. Therefore, N can exert a variety of negative effects on plant defense patterns [10, 12, 20]. High N availability in maize significantly increased N and total protein content, subse-quently, leaf-chewing S. exigua larvae showed higher performance and preferentially fed on high-N maize [12]. This and our results are in accordance with the resource availability hypothesis, which predicts that plants in resource-rich environments have lower levels of defense and encounter higher levels of herbivory than plants in resource-poor environ-ments [20, 39]. Although S. frugiperda larvae demonstrated a higher preference for N2 maize, their corresponding RGR was not increased remarkably. Probably, insect had to allocate some energies taken up from maize leaves to detoxify secondary metabolites, which was consist with the increased the production of total phenolics and DIMBOA along with N levels (Fig. 2). Xu et al. reported that increased N fertilization in maize en-hanced the total phenolics content and ultimately decreased the RGR of O. furnacalis [13]. However, plant nutrition status was positively closely associated with leaf area fed by S. frugiperda larvae than the production of secondary metabolites (Fig. 3f), indicating that N supply had the negative effects of N supply on maize defense and maize nutrient status not secondary metabolites drove herbivory without any induced treatments [40].”

Thank you again for your valuable comments concerning our manuscript. We hope this revision is acceptable and readable.

Reviewer 2 Report

This paper describes a complex series of experiments documenting responses of maize plants to N-fertilisation and herbivore damage. Weaknesses include old-fashioned methods (Bradford analysis, total phenolics) and a small sample size (N=4). However, I do not feel this prevents publication.  The paper does need to be situated more clearly in the current research on maize (notably that from Turling's lab), showing clearly what is already known about maize defense (a great deal!) and what this paper contributes that is novel.  This is not the first paper examining N fertilization and defenses in maize, so what does it add to what we already know?  In addition, the writing should be more concise (notably in the results and discussion sections), omitting redundant explanations of results and focusing on maize literature. It is not possible to cite all the examples of N-fertilization and insect behavior, so focus on those that are the most relevant.

Specific comments:

Title not very informative

l.18 'negative bottom-up effects' is confusing: skip straight to saying that it enhanced nutrient content of leaves and feeding by S frugiperda.

l. 23 It's confusing to say that supplementation triggered defense in an N-dependent manner. Would it be more accurate to say that N fertilization both increases leaf nutrient content and enhances induced defense (when coupled with herbivory or JA) and therefore effects on herbivores are mixed?

l.48-50: not very informative. N can affect host choice and insect nutrition and secondary metabolites.

l.50-54: there are hundreds of studies on N fertilization on plant-insect interactions. Why cite these? Is there a recent review? Or literature on mechanisms of plant physiology?

Focus on two main effects: N on nutrition and N on defensive compounds (including VOCs).

l.65-70: this is confusing. Is this all on tomato? Presumably defensive compounds are induced by herbivory, but the extent to which induction occurs depends on N available to the plant. Would this be an accurate statement?

l.75-77: Please summarize the results of some of these studies on Maize (in China and elsewhere, including the substantial body of work from Turling's lab). What have we learned about induced responses in maize? and how they are modulated by Nitrogen? this would be more relevant that the details of compounds produced by tomato.

Methods:

replication N=4. This is quite low.  Confirm that it is the case for all analyses.

l. 439: not clear. What was done with the 6 random plants? Were all 36 plants destructively sampled to obtain biomass measures?

3 larvae were placed on 4 plants per treatment for herbivory treatment. Were these the same larvae used for growth measurements? If so, this means 10 of 12 larvae were transfered to rearing.  How were leaves obtained? From the same plants as above?  Were the same leaves kept for the 7 day period or were they replaced?

Similarly, it's not clear if where the plants and the insects used for the feeding behavior experiment and the olfactometer experiment came from. Were different individuals used in different experiments? In which case, to which experiment does the experimental design section refer?

l.571: PCR for what? please clarify in the title that the goal was to measure specific enzymes and why these enzymes were chosen. Title in l. 231 is more informative.

Fig. 2; x-axis legend not clear. Please describe in caption what N0CK etc means. Also, the role of panels e & f is not clear: are they the same data? It seems not, but the presentation of the two panels is confusing.  The pictures for the Sf and JA treatments do not appear correctly placed on the figure on my copy of the MS.

Results:

l.114: This sentence is not clear, please rephrase. Do you mean that in the induced treatments (either with herbivory or JA) attraction increased on N1 plants, but decreased on N2 plants?

l.138: If the treatments overall did not affect soluble sugars, then the analysis should stop there. Individual significant effects should not be considered if the overall analysis shows no significance. A similar reasoning applies to other analyses.

This is a complex experiment with many results. Nonetheless, I feel the results section could be written in a more concise and synthetic fashion to make it easier for the reader to follow.  Focus only on significant results and avoid redundant text.

Discussion:

l.275: N supplementation did increase protein and amino acids but not sugars. This is confusing as it seems contradictory with results section.

Here too, there are dozens of papers showing how N supplementation changes plant nutrient content for herbivores: focus on work on maize and on a recent review. Please focus on maize literature throughout the text.

l.290-293: this is confusing.  Resuls suggest that S.f. prefer fertilized maize initially, probably due to increased available amino acids and/or protein. But post-damage, fertilized maize becomes less preferred due to induced defenses. It seems likely that amino acids and protein are low because they are taken up to produce defenses.  How does this play out in the field? Presumably preference for fertilized maize only lasts until induced defenses become expressed, which takes longer than the time-frame of your experiment.

l.295-308: it seems that induced defenses strong enough to influence S.f. require high levels of fertilization.  l.302-304: saying that low levels of fertilizer 'weakened defenses' appears misleading. Instead, it seems more likely that it increased nutritional quality without strengthening induced defenses much, making plants more attractive to the insect.

Author Response

Response to Reviewer 2 Comments

Point 1: This paper describes a complex series of experiments documenting responses of maize plants to N-fertilisation and herbivore damage. Weaknesses include old-fashioned methods (Bradford analysis, total phenolics) and a small sample size (N=4). However, I do not feel this prevents publication. The paper does need to be situated more clearly in the current research on maize (notably that from Turling's lab), showing clearly what is already known about maize defense (a great deal!) and what this paper contributes that is novel. This is not the first paper examining N fertilization and defenses in maize, so what does it add to what we already know?  In addition, the writing should be more concise (notably in the results and discussion sections), omitting redundant explanations of results and focusing on maize literature. It is not possible to cite all the examples of N-fertilization and insect behavior, so focus on those that are the most relevant.

Response 1: Dear Reviewer, We greatly appreciate your patient and valuable comments. We have thoroughly considered all the comments and improved our manuscript accordingly. In the revision, we described the deatail replications in every measurements; we also revised the results and discussion as well as references and deleted some redundant sentences and references; we also try to focus on literatures related with maize defense.

Point 2: Title not very informative.

Response 2: We have revised the title as “Effects of Nitrogen Supply on Induced Defense in Maize (Zea mays) Against Fall Armyworm (Spodoptera frugiperda)”.

Point 3: l.18 'negative bottom-up effects' is confusing: skip straight to saying that it enhanced nutrient content of leaves and feeding by S frugiperda.

Response 3: We have revised this sentence as “In the absence of herbivore attack or exogenous jasmonic acid (JA) application, N supply increased plant biomass and enhanced maize nutrient (soluble sugar and amino acid) contents and leaf area fed by S. frugiperda (the feeding leaf area of S. frugiperda larvae in maize supplemented with 52.2 and 156.6 mg/kg N was 4.08 and 3.83 times that of the control, respectively).”

Point 4: l. 23 It's confusing to say that supplementation triggered defense in an N-dependent manner. Would it be more accurate to say that N fertilization both increases leaf nutrient content and enhances induced defense (when coupled with herbivory or JA) and therefore effects on herbivores are mixed?

Response 4: Thanks for your suggestions. We have revised this sentence as “When coupled with herbivore attack or JA application, maize supplemented with 52.2 mg/kg N showed an increased susceptibility to pests, while the maize supplemented with 156.6 mg/kg N showed an improved defense against pests. The changes in the levels of nutrients, and the emissions of volatile organic compounds (VOCs) caused by N supply could explain the above opposite induced defense in maize.”

Point 5: l.48-50: not very informative. N can affect host choice and insect nutrition and secondary metabolites.

Response 5: In the reference (Lu, Z.; Yu, X.; Heong, K.; Hu, C.; Effect of nitrogen fertilizer on herbivores and its stimulation to major insect pests in rice. Rice Sci. 2007, 14, 56-66.), authors reported that N fertilizer application rarely affects insect directly, however, it can alter or change morphological, biochemical and physiological characters of host plants and improve nutritional conditions for herbivores. Therefore, N fertilizer application affects the ecological fitness of herbivores via bottom-up effects (from plants to insects). The ecological fitness of herbivores included selection of host plants, survival, growth, development, fecundity, and population dynamics.

In the revision, we have revised this sentence as “N also impacts the ability of plants to cope with biotic stress, including herbivore attack [18]. N fertilizer may not affect insect biology directly but change the host-plant morphology, biochemistry and physiology, which can improve nutritional conditions for herbivores [19]. Therefore, N supply has bottom-up effects on the ecological fitness of herbivores, including selection of host plants, survival, growth, development, fecundity, and population dynamics [10, 20]”.

Point 6: l.50-54: there are hundreds of studies on N fertilization on plant-insect interactions. Why cite these? Is there a recent review? Or literature on mechanisms of plant physiology?

Response 6: We collected and read the numbers of literatures on N fertilization and plant-insect interactions. Sorry for limited knowledge, we did not find a recent review on effects of N fertilization on plant-insect interactions. There was one paper published in 2010 (Effects of nitrogen fertilization on tritrophic interactions, Arthropod-Plant Interactions, 2010, 4: 81–94).

There were only 4 literatures on N fertilization and maize-insect interactions: 1) Corn defense responses to nitrogen availability and subsequent performance and feeding preferences of beet armyworm (Lepidoptera: Noctuidae) (Field and Forage Crops, 2013, 13: 1240-1249). 2) Effects of elevated CO2 and increased N fertilization on plant secondary metabolites and chewing insect fitness (Front. Plant Sci. 2019, 10: 739). 3) Nitrogen deficiency increases volicitin-induced volatile emission, jasmonic acid accumulation, and ethylene sensitivity in maize (Plant Physiology, 2003, 133, 295–306). 4) Effect of slag and nitrogen fertilizer on the damage of Asian corn borer to field corn (Mem Coll Agric, Natl Taiwan Univ, 1994, 34(1): 45-53, In Chinese).

According to your comments, we focused on maize defense. So, we revised this paragraph as “N also impacts the ability of plants to cope with biotic stress, including herbivore attack [18]. N fertilizer may not affect insect biology directly but change the host-plant morphology, biochemistry and physiology, which can improve nutritional conditions for herbivores [19]. Therefore, N supply has bottom-up effects on the ecological fitness of herbivores, including selection of host plants, survival, growth, development, fecundity, and population dynamics [10, 20]. For example, N application caused rice susceptible to Nilaparvata lugens [19]; the increased N availability in maize leads to an increased preference for beet armyworm Spodoptera exigua [12] and the number of egg masses deposited by Ostrinia furnacalis [20].”

Point 7: l.65-70: this is confusing. Is this all on tomato? Presumably defensive compounds are induced by herbivory, but the extent to which induction occurs depends on N available to the plant. Would this be an accurate statement?

Response 7: No, this is not all on the tomato. We have checked the reference and revised this sentence as “Under low N conditions, the cotton attack by Spodoptera exigua significantly increased the total amount and types of VOCs, such as (Z)-3-hexenyl, (E)-2-hexenal, β-caryophyllene, (E)-β-farnesene, α-humulene and γ-bisabolene [27, 31-32]. N deficiency increases volicitin induced volatile emission in maize [31]. However, an increase in N supply significantly reduced S. exigua larval-induced VOC concentrations in cotton [27, 31-32].”

Point 8: l.75-77: Please summarize the results of some of these studies on Maize (in China and elsewhere, including the substantial body of work from Turling's lab). What have we learned about induced responses in maize? and how they are modulated by Nitrogen? this would be more relevant that the details of compounds produced by tomato.

Response 8: We collected and read literatures (reviews) on maze defense with against herbivores (in China and elsewhere, including the substantial body of work from Turling's lab). We also focus on N fertilization and maize-insect interactions, as well as Point 6. So, we revised this sentence as “For approximately three decades, numerous laboratories have investigated the defense systems of maize against herbivores, including secondary metabolites [35-37], phytohormones [38], and elicitation of VOCs [4]. However, effects of environmental factors (e.g., N supply) on maize induced defense patterns and its consequence for insect herbivores are rarely studied [12-13].”

Point 9: replication N=4. This is quite low.  Confirm that it is the case for all analyses.

Response 9: There were 4 or more than 4 replications for every measurement. For plant aboveground biomass, 6 replications for each treatment combination. For growht of S. frugiperda, 10*3 replications for each treatment combination. For feeding behaviour of S. frugiperda, 20 replications for each treatment combination. For total protein, soluble sugar, amino acids, phytohormones, expression of genes and VOCs, 4 replications for each treatment combination. For secondary metabolites, 6 replications for each treatment combination. Therefore, we have revised the section “materials and methods” and wrote the number of replications for every meansurement.

Point 10: l. 439: not clear. What was done with the 6 random plants? Were all 36 plants destructively sampled to obtain biomass measures?

Response 10: In this measure, we planted 10 maize plants for each treatment combination and 6 maize plants were randomly selected for this measurement. In 4.2. Experimental design and conditions, we have revised last sentence as “In total, we constructed 9 treatment combinations and more than 4 replicates for each treatment combination in every measurement.”

Point 11: 3 larvae were placed on 4 plants per treatment for herbivory treatment. Were these the same larvae used for growth measurements? If so, this means 10 of 12 larvae were transfered to rearing.  How were leaves obtained? From the same plants as above?  Were the same leaves kept for the 7 day period or were they replaced?

Response 11: Three larvae were placed on one plant for herbivory treatment, more than 4 plants per N level. These were different larvae used for growth meansurements. For the larval growth measurement, we fed other 30 of the 3rd instar larvae with fresh leaves of maize from each treatment combination (herbivore attack or JA application, different N levels). These maize plants were different from above. We cut fresh leaves from plants and replace them to feed larvae daily.

Therefore, we also revised this section as “After herbivore attack or JA application for 12 h, we fed 3rd instar larvae with fresh leaves of maize growing in soil supplemented with different N levels. After measuring the initial weight of the larvae, 10 larvae from every treatment combination were transferred to plastic boxes (4 cm diameter) to rear individually with enough fresh leaves daily; each bi-oassay was performed in triplicate. We measured the weight of larvae after 7 d. The rela-tive growth rate (RGR) was calculated as RGR (%) = (Wn-W0)/(W0×T) × 100% [72], where W0 was the initial weight of S. frugiperda, Wn was the fresh weight at the check day, and T (d) was the duration of the experimental period.”

Point 12: Similarly, it's not clear if where the plants and the insects used for the feeding behavior experiment and the olfactometer experiment came from. Were different individuals used in different experiments? In which case, to which experiment does the experimental design section refer?

Response 12: Different individuals were used in different experiments. The experimental design mainly showed that two factors: different N levels (N0, N1 and N2) and plants with different treatments: untreated (control, CK), S. frugiperda attack (SF), and jasmonic acid application (JA). For all measure (4.3. 4.4, 4.5, 4.6, 4.7, 4.8, 4.9), samples were collected from each treatment combination (herbivore attack or JA application, different N levels).

In the revision, we rewrote last sentence in the section 4.2 as “In total, we constructed 9 treatment combinations and more than 4 replicates for each treatment combination in every measurement.”

Point 13: l.571: PCR for what? please clarify in the title that the goal was to measure specific enzymes and why these enzymes were chosen. Title in l. 231 is more informative.

Response 13: PCR for expression of maize defensive genes. Thanks for your suggestions. We have revised “Quantitative real-time PCR analysis” as “Quantitative real-time PCR analysis of maize defensive genes”.

Point 14: Fig. 2; x-axis legend not clear. Please describe in caption what N0CK etc means. Also, the role of panels e & f is not clear: are they the same data? It seems not, but the presentation of the two panels is confusing.  The pictures for the Sf and JA treatments do not appear correctly placed on the figure on my copy of the MS.

Response 14: According to your commenets, we reploted this figure, rewrote the title and description of this figure (Figure in the attachment).

Figure 2. Effects of N supply on preference of S. frugiperda larvae towards maize under different induced treatments. (a), Relative growth rate (RGR) of S. frugiperda larvae; (b), feeding leaf area (cm2) by S. frugiperda larvae; (c), behavioral response (number) of S. frugiperda towards maize plants from different N levels under the same induced treatments; (d), behavioral response (number) of S. frugiperda towards maize plants from different induced treatments under the same N levels.

Point 15: l.114: This sentence is not clear, please rephrase. Do you mean that in the induced treatments (either with herbivory or JA) attraction increased on N1 plants, but decreased on N2 plants?

Response 15: We revised this paragaraph as “In the absence of induced treatments, S. frugiperda preferred maize plants growing in N2 soil (Fig. 2c). Herbivore attack enhanced N1 maize to attract insects, while the choice of S. frugiperda to N2 maize was significantly decreased (Fig. 2c). After JA application, the choice of N2 maize by S. frugiperda was also decreased (Fig. 2c). Regardless of whether the maize plants were growing in N0, N1 or N2, induced treatments enhanced the ability of the plants to repel S. frugiperda larvae (Fig. 2d).”

Point 16: l.138: If the treatments overall did not affect soluble sugars, then the analysis should stop there. Individual significant effects should not be considered if the overall analysis shows no significance. A similar reasoning applies to other analyses.

Response 16: We rechecked and analyzed the original data again. For soluble sugars, there was a exception data in maize growing in N1 soil with JA apllication. Therefore, we removed this exception data and conducted GLM analysis, one-way ANOVA, T-test and Pearson correlation again. In the revision, we also wrote this paragraph as “N levels, individually and in combination with induced treatments, affected the con-tent of soluble sugar and amino acids (Fig. 3b, 3c). In the absence of induced treatments, N supply significantly increased the contents of soluble sugar and amino acids (Fig. 3b). However, the contents of soluble sugars in N2 maize with induced treatments were sig-nificantly lower than those of the corresponding N0 and N1 maize (Fig. 3b). Compared to untreated plants, N2 maize with induced treatments significantly had reduced contents of soluble sugar (Fig. 3b). N1 maize with induced treatments showed significantly increased amino acid contents. In contrast, N2 maize with induced treatments had significantly re-duced amino acid contents (Fig. 3c).”

Moreover, we also checked the original data of all measurements again. Thanks for your careful comments.

Point 17: This is a complex experiment with many results. Nonetheless, I feel the results section could be written in a more concise and synthetic fashion to make it easier for the reader to follow.  Focus only on significant results and avoid redundant text.

Response 17: Thanks for your suggestion. We revised this section and redundant sentences were deleted. We hope this revision is acceptable and readable. Thank you again.

Point 18: l.275: N supplementation did increase protein and amino acids but not sugars. This is confusing as it seems contradictory with results section.

Response 18: Without any induced treatments, N supplementation did increase soluble protein, soluble sugars and amino acids. It is not contradictory with results.

Point 19: Here too, there are dozens of papers showing how N supplementation changes plant nutrient content for herbivores: focus on work on maize and on a recent review. Please focus on maize literature throughout the text.

Response 19: Thanks for your suggestions. We try to focus on maize literature throughout the text according to your comments.

We revised this paragraph as “The present study demonstrated the negative effects of N supply on maize defense against S. frugiperda larvae, in which higher N levels increased plant biomass (Fig. 1) and enhanced nutrient contents (soluble protein, soluble sugar and amino acids) of maize plants without any treatments (herbivore attack or JA application) (Fig. 3), which im-proved the leaf quality and consequently S. frugiperda larvae fed more maize leaves (Fig. 2b). The plants growing in N increased the nutritional quality (biosynthesis or accumula-tion of proteins, free amino acid and sugars) that might have attracted insects [10, 18, 20]. Therefore, N can exert a variety of negative effects on plant defense patterns [10, 12, 20]. High N availability in maize significantly increased N and total protein content, subse-quently, leaf-chewing S. exigua larvae showed higher performance and preferentially fed on high-N maize [12]. This and our results are in accordance with the resource availability hypothesis, which predicts that plants in resource-rich environments have lower levels of defense and encounter higher levels of herbivory than plants in resource-poor environ-ments [20, 39]. Although S. frugiperda larvae demonstrated a higher preference for N2 maize, their corresponding RGR was not increased remarkably. Probably, insect had to allocate some energies taken up from maize leaves to detoxify secondary metabolites, which was consist with the increased the production of total phenolics and DIMBOA along with N levels (Fig. 2). Xu et al. reported that increased N fertilization in maize en-hanced the total phenolics content and ultimately decreased the RGR of O. furnacalis [13]. However, plant nutrition status was positively closely associated with leaf area fed by S. frugiperda larvae than the production of secondary metabolites (Fig. 3f), indicating that N supply had the negative effects of N supply on maize defense and maize nutrient status not secondary metabolites drove herbivory without any induced treatments [40].”

Point 20: l.290-293: this is confusing.  Resuls suggest that S.f. prefer fertilized maize initially, probably due to increased available amino acids and/or protein. But post-damage, fertilized maize becomes less preferred due to induced defenses. It seems likely that amino acids and protein are low because they are taken up to produce defenses. How does this play out in the field? Presumably preference for fertilized maize only lasts until induced defenses become expressed, which takes longer than the time-frame of your experiment.

Response 20: In our study, we used the feeding leaf area (Leaf discs) as an important index to estimate maize defense. Less leaf area fed by S.f. indicated stronger defense of maize. Without any induced treatments, higher N levels led to more leaf area fed by S.f., indicating weaker defense of maize. Although S.f. fed more higher N maize leaves, the relative rate growth of S.f. was not significantly increased. This indicated that S.f. aalocated energy from leaves to resist secondary metabolites (the increased total phenolics and DIMBOA along with N levels). In the revision, we revised this sentence as “Although S. frugiperda larvae demonstrated a higher preference for N2 maize, their corre-sponding RGR was not increased remarkably. Probably, insect had to allocate some energies taken up from maize leaves to detoxify secondary metabolites, which was consist with the increased the production of total phenolics and DIMBOA along with N levels (Fig. 2). Xu et al. reported that increased N fertilization in maize enhanced the total phenolics content and ultimately decreased the RGR of O. furnacalis [13]. However, plant nutrition status was positively closely associated with leaf area fed by S. frugiperda larvae than the production of secondary metabolites (Fig. 3f), indicating that N supply had the negative effects of N supply on maize defense and maize nutrient status not secondary metabolites drove herbivory without any induced treatments [40].”

Thanks for your suggestions. We continue to investigate effects of N supply on maize defense with herbivorous insect in the lab and the field. More N levels and longer time, priming chemicals (JA or SA) and pest natural enemies were conducted in the furture study.

Point 21: l.295-308: it seems that induced defenses strong enough to influence S.f. require high levels of fertilization.  l.302-304: saying that low levels of fertilizer 'weakened defenses' appears misleading. Instead, it seems more likely that it increased nutritional quality without strengthening induced defenses much, making plants more attractive to the insect.

Response 21: Thanks for your suggestions. We revised this sentence as “The 156.6 mg/kg N supplementation enhanced the defense of maize plants against pests, while the 52.2 mg/kg N supplementation increased nutritional quality without strength-ening induced defenses much, making plants more attractive to the insect.”

Thank you again for your valuable comments concerning our manuscript. We hope this revision is acceptable and readable.

Reviewer 3 Report

The authors showed good work on how nitrogen supply affects the induced defense of plants. they investigated the impacts of N supply on the defense induced in maize (Zea mays) against the fall armyworm (Spodoptera frugiperda). In the absence of herbivore attack or exogenous jasmonic acid (JA) application. They found higher N levels increased plant biomass and enhanced maize nutrient (soluble sugar and amino acid) contents to improve leaf quality as a pest food resource (the feeding leaf area of S. frugiperda larvae in maize supplemented with 52.2 and 156.6 mg/kg N was 4.08 and 3.83 times that of the control, respectively).

I have some comments 

- In material and methods... line 418 it is not (or) but and. 

- Nitrogen levels are treatments too, so must be changed second factor (CK, SF, and JA) to another name, so that treatments include both Nitrogen levels and [(control, CK),  S. frugiperda attack (SF), and jasmonic acid application (JA)].

-  What was the number of plants grown in each pot?

- What was the number of plants that took them to read from each replicate?

- Figure 1 needs further clarification in treatments titles. 

-  How many samples (reads) were used in the T-test? 

Author Response

Response to Reviewer 3 Comments

Point 1: In material and methods... line 418 it is not (or) but and..

Response 1: Dear Reviewer, We greatly appreciate your patient and valuable comments. We have thoroughly considered all the comments and improved our manuscript accordingly. We have revised this sentence as “We conducted a 2×2 full factorial experiment, in which we manipulated N supplemented soil, S. frugiperda attack and jasmonic acid (JA) application as the two main factors.”

Point 2: Nitrogen levels are treatments too, so must be changed second factor (CK, SF, and JA) to another name, so that treatments include both Nitrogen levels and [(control, CK),  S. frugiperda attack (SF), and jasmonic acid application (JA)].

Response 2: You are right, we have 9 treatment combinations. In the revision, we have revised “three treatments” as “the second factor”. The details were “We conducted a 2×2 full factorial experiment, in which we manipulated N supple-mented soil, S. frugiperda attack and jasmonic acid (JA) application as the two main factors. Three levels of N were added to the soil for the growing maize plants, namely, 1) no N ad-dition (N0), 2) addition of 52.2 mg/kg N (N1) and 3) addition of 156.6 mg/kg N (N2). The second factor was used: 1) untreated (control, CK), 2) S. frugiperda attack (SF), and 3) jasmonic acid application (JA). In total, we constructed 9 treatment combinations and more than 4 replicates for each treatment combination in every measurement.”

Point 3: What was the number of plants grown in each pot?.

Response 3: One plant was grown in each pot.

Point 4: What was the number of plants that took them to read from each replicate?

Response 4: One plant was took for each replicate. More than 4 replicates per each treatment combination were conducted in every measurement.

Point 5: Figure 1 needs further clarification in treatments titles.

Response 5: We have replotted Figure 1 (Figure in attachment) and written the titles in the revision.

Figure 1. Effects of Nitrogen (N) supply on the aboveground biomass of maize under different induced treatments (herbivore attack or JA application). Generalized linear models (GLM) results of testing for N level and induced treatment effects are shown (n.s., not significant; *P<0.05; **P<0.01; ***P<0.001); different lowercase, uppercase and italic lowercase letters above the bars indicate significant differences among different N levels (supplemented with 0, 52.2, 156.6 mg/kg N, denoted as N0, N1, N2) under the same induced treatments at P<0.05 (one-way analysis of variance (ANOVA)). The asterisk “*” above the bars indicates significant differences between maize with and without S. frugiperda attack; “#” above the bars indicates significant differences between maize with and without JA application by T-test. The same below.

Point 6: How many samples (reads) were used in the T-test?

Response 6: More than 4 samples for every treatment combination were used in the T-test.

Thank you again for your valuable comments concerning our manuscript. We hope this revision is acceptable and readable.
